# A Normative Model Representing Autistic Individuals Amidst Autism Spectrum Phenotypic Heterogeneity

**DOI:** 10.3390/brainsci14121254

**Published:** 2024-12-14

**Authors:** Joana Portolese, Catarina Santos Gomes, Vinicius Daguano Gastaldi, Cristiane Silvestre Paula, Sheila C. Caetano, Daniela Bordini, Décio Brunoni, Jair de Jesus Mari, Ricardo Z. N. Vêncio, Helena Brentani

**Affiliations:** 1Laboratório de Psicopatologia e Terapêutica Psiquiátrica (LIM23), Faculdade de Medicina FMUSP, Universidade de São Paulo, Sao Paulo 05403-010, SP, Brazil; joanaportolese@gmail.com (J.P.); catarinagomes@usp.br (C.S.G.); vinidg@gmail.com (V.D.G.); 2Departamento & Instituto de Psiquiatria, Faculdade de Medicina FMUSP, Universidade de São Paulo, Sao Paulo 05403-010, SP, Brazil; 3Pós-Graduação em Ciências do Desenvolvimento Humano, Universidade Presbiteriana Mackenzie, São Paulo 01302-907, SP, Brazil; csilvestrep09@gmail.com (C.S.P.); debruno46@gmail.com (D.B.); 4Social Cognition Clinic—TEAMM, Department of Psychiatry, Universidade Federal de São Paulo (UNIFESP), São Paulo 04021-001, SP, Brazil; danibordini4@gmail.com; 5Department of Psychiatry, Universidade Federal de São Paulo (UNIFESP), São Paulo 04021-001, SP, Brazil; sheilaccaetano@gmail.com (S.C.C.); jamari17@gmail.com (J.d.J.M.); 6Department of Computing and Mathematics FFCLRP, Universidade de São Paulo (USP), Ribeirão Preto 14040-901, SP, Brazil; rvencio@usp.br

**Keywords:** autism spectrum disorder, phenotypic heterogeneity, normative modeling, principal component analysis

## Abstract

Background: Currently, there is a need for approaches to understand and manage the multidimensional autism spectrum and quantify its heterogeneity. The diagnosis is based on behaviors observed in two key dimensions, social communication and repetitive, restricted behaviors, alongside the identification of required support levels. However, it is now recognized that additional modifiers, such as language abilities, IQ, and comorbidities, are essential for a more comprehensive assessment of the complex clinical presentations and clinical trajectories in autistic individuals. Different approaches have been used to identify autism subgroups based on the genetic and clinical heterogeneity, recognizing the importance of autistic behaviors and the assessment of modifiers. While valuable, these methods are limited in their ability to evaluate a specific individual in relation to a normative reference sample of autistic individuals. A quantitative score based on axes of phenotypic variability could be useful to compare individuals, evaluate the homogeneity of subgroups, and follow trajectories of an individual or a specific group. Here we propose an approach by (i) combining measures of phenotype variability that contribute to clinical presentation and could impact different trajectories in autistic persons and (ii) using it with normative modeling to assess the clinical heterogeneity of a specific individual. Methods: Using phenotypic data available in a comprehensive reference sample, the Simons Simplex Collection (*n* = 2744 individuals), we performed principal component analysis (PCA) to find components of phenotypic variability. Features that contribute to clinical heterogeneity and could impact trajectories in autistic people were assessed by the Autism Diagnostic Interview-Revised (ADI-R), Vineland Adaptive Behavior Scales (VABS) and the Child Behavior Checklist (CBCL). Cognitive assessment was estimated by the Total Intelligence Quotient (IQ). Results: Three PCs embedded 72% of the normative sample variance. PCA-projected dimensions supported normative modeling where a multivariate normal distribution was used to calculate percentiles. A Multidimensional General Functionality Score (MGFS) to evaluate new prospective single subjects was developed based on percentiles. Conclusions: Our approach proposes a basis for comparing individuals, or one individual at two or more times and evaluating homogeneity in phenotypic clinical presentation and possibly guides research sample selection for clinical trials.

## 1. Introduction

Autism is multifactorial, with a complex genetic architecture and high variability in phenotypic presentation [1]. Different studies [2,3,4,5], mainly using factor analysis, contributed to the Diagnostic and Statistical Manual of Mental Disorders, 5th edition (DSM-5) [6], criteria based on two dimensions of autistic behaviors: social communication and restricted and repetitive behaviors. There is substantial variability in autistic behaviors among individuals [7,8,9], and the levels of required support can vary accordingly [10]. Established instruments to estimate levels of autistic behaviors, such as the Childhood Autism Rating Scales (CARSs) [11] and the ADOS-2 Calibrated Severity Scores (CSSs) have been used in clinical practice. The total scores in the Autism Diagnostic Observation Schedule (ADOS), a golden standard instrument for diagnosis, were revised to provide a continuous measure of severity less influenced by child characteristics, such as age and language, yielding the ADOS-2 Calibrated Severity Scores (CSSs) [12,13,14]. The CSS produces a symptom severity rating that is standardized against individuals of the same age and language abilities.

The important role of factors beyond autistic behaviors, such as language level, IQ, and comorbidities, has been recognized to contribute to phenotypic heterogeneity and the DSM-5 recommends using these modifiers to better cover the heterogeneity [15,16,17,18]. The presence of emotional/behavioral problems such as disruptive behaviors [19] is relatively common, including outbursts of anger, irritability, opposing behavior, and aggression [20], and contributes to the phenotypic heterogeneity [21,22]. It also contributes to the needed support levels [23]. These additional challenges, such as intellectual difficulties, limited expressive and/or receptive language, and anxiety disorders, can have a significant impact on the daily lives of autistic individuals, both in terms of their adaptive functioning and their sense of well-being [24]. Moreover, they are recognized as important features contributing to different trajectories in autistic persons [25,26,27,28]. Furthermore, autistic behaviors and co-occurring conditions may interact across development, each influencing the trajectory of the other over time.

The phenotypic heterogeneity brings difficulties, both in clinical practice and in the research area [29,30]. Different studies have used IQ, functionality, emotional/behavioral problems, and other phenotypic measures along with autistic behaviors to identify subgroups and better characterize clinical heterogeneity [25,26,27,28] Cluster analysis is suitable for grouping subjects based on patterns in the data in order to increase the similarity within the clusters. Therefore, cluster analysis groups individuals with autistic behaviors according to their similarity, for a wide range of variables [31,32,33]. It is clear that subgroups exist within the autistic behavior spectrum and probably various etiological risk factors combine in different ways to better explain the heterogeneity [34,35,36]. However, these studies often do not focus on individual patients or compare individuals in a standardized manner, an important task in everyday clinical practice. To understand how autism affects a person’s functioning, well-being, and daily life, we must understand how the different aspects of autism and their modifiers interact with each other at specific timepoints and across development. It is important to assess how autism impacts a person’s life in a more comprehensive way that addresses the autistic behaviors as well as other influential aspects of an autistic person’s life [24]. This multidimensional perspective can be meaningful for clinical assessment, i.e., identifying needs, planning interventions, assigning support, and creating future goals based on their overall level of functioning.

Recently, the promising approach of “normative modeling” has been used to characterize the heterogeneity of psychiatric disorders [37]. This method employs a probabilistic model to determine the joint distribution and variation of symptoms and biological characteristics, accounting for relevant covariates. Using normative representations, the degree to which individuals deviate from reference population ranges can be assessed in a personalized fashion.

The studies that carried out the normative modeling approach to better characterize psychiatric disorders used healthy individuals to create normative population ranges (reviewed in [37]). However, it is also possible to create a map of normative ranges to understand the heterogeneity of a specific clinical presentation, e.g., in autistic individuals. In this context, a patient can be evaluated on the normative map, giving clinicians an idea of relative positions, and orienting not only the comparison among individuals, finding more homogeneous subgroups, but also individual or subgroups changes over time.

Translating this multidimensional perspective into a measurable and formal definition that captures the combined influences of autism and comorbidity features on an individual’s life, as well as providing a detailed characterization for research, is still a challenge [24]. In this work, we propose a two-step integrated approach to measure components of phenotype variability that can impact an autistic person’s life in a more comprehensive way and use normative modeling to allow the comparison of phenotypic variability at the individual level. As a large reference sample, we used the Simons Foundation Autism Research Initiative (SFARI). We used principal component analysis on measures of cognition, adaptive function, the regulation of emotional behaviors, social communication, and repetitive and/or stereotyped behaviors, as a dimension reduction technique to find components of phenotypic variability that take into account autistic behaviors traits and other co-occurring characteristics that could influence individuals’ autonomy and trajectories over time. As each principal component is orthogonal to the others, all together will create a dimensional space of heterogeneity variation. We propose a Multidimensional General Functionality Score (MGFS) that represents a value in the space, for each individual, taking into account information from all principal components. We would expect that the proposed Multidimensional General Functionality Score (MGFS) would not correlate strongly with the ADOS-2 Calibrated Severity Scores (CSSs), which focus solely on autistic core-diagnostic behaviors. Finally, the proposed approach was applied to an autistic behavior sub-sample of a previously published study [38] demonstrating how the proposed approach could contribute to evaluating the phenotypic homogeneity of sub-samples, necessary for clinical trials.

## 2. Materials and Methods

### 2.1. Participants

To investigate components of phenotypic variability in ASD, we used 2744 probands from the Simons Simplex Collection (SSC) sample (version 15.0) of the Simons Foundation Autism Research Initiative (SFARI) (Fischbach & Lord, 2010) [39]. The sample included 2359 males and 385 females from 2 to 18 years, selected based on the total completeness of the target scales and IQ score questionnaires.

We also used, as a case study, 27 children diagnosed with ASD from a previously published randomized clinical trial of a video modeling parenting training [38]. The inclusion criteria were as follows: patients with ASD diagnosed using ADI-R, ages between 3 and 7 years old, and IQ between 50 and 75. Exclusion criteria were the presence of a known genetic syndrome evaluated by a clinical geneticist and individuals without the Risk Scores evaluation. The patients had measurements of functionality and socio-emotional skills assessed twice in an interval of 8 months. Appendix A describe the scales/questionnaires’ mean and standard deviation scores, relative to the reference and case samples, respectively.

### 2.2. Assessment Tools

The Autism Diagnostic Interview-Revised (ADI-R), Vineland Adaptive Behavior Scales (VABS), and Child Behavior Checklist (CBCL) were used for assessing various aspects of autistic-related behaviors in individuals with autism. The ADI-R [40] is a structured caregiver interview focused on identifying autism-related behaviors in communication, social interaction, and restricted or repetitive behaviors. The VABS [41] evaluates adaptive behaviors, covering daily living skills, socialization, motor skills, and communication abilities, thus providing insights into an individual’s functional capabilities. The CBCL [42], is a caregiver questionnaire that assesses a broad range of emotional and behavioral issues, including internalizing and externalizing problems, and is useful for identifying co-occurring behavioral challenges. Together, these scales offer a comprehensive understanding of the behavioral profile, adaptive functioning, and specific autistic traits. We also considered cognitive measurements estimated by the Total Intelligence Quotient (IQ) using both the Full-Scale Deviation IQ data whenever available, and alternatively, the Full-Scale Ratio IQ. In the case study sample, children were evaluated using the Snijders–Oomen Nonverbal Intelligence Test—Revised (SON-R 2½–7) [43].

### 2.3. Principal Component Analysis

Principal component analysis (PCA) is a dimensionality reduction technique used to extract significant features, (principal components), from a large set of variables in a dataset. Each principal component is a linear combination of the original variables, and contains no redundant information, ensuring each new variable generated holds the maximum variance.

In this study, PCA was applied to capture the main sources of phenotypic variability within the normative reference dataset (SFARI). Input variables included the VABS domains of Socialization, Communication, and Daily Living Skills; the ADI-R domains of Socialization, Communication, and Restricted and Repetitive Behaviors; total IQ; and t-scores for CBCL Internalizing and Externalizing Problems. To ensure comparability among variables, scores were standardized into z-scores. PCA was then performed using the ‘prcomp’ function in R, with default parameters except for setting ‘scale = TRUE’ to normalize the data.

Principal components (PCs) with an eigenvalue greater or equal to 1.0 hold the same amount of information as a single original variable. Therefore, the principal components (PCs) with an eigenvalue greater than 1.0, or those that together explained up to 70% of the sample’s variability, were selected for further analysis [44]. To assign interpretability to each PC, we consider the original variables presenting absolute correlations greater than 0.5 with each PC as relevant. To assess the stability of the PCA-based phenotypic structure, we repeated the analysis on varying subsets of the SSC sample, selecting portions ranging from 30% up to 80% of the total sample.

The principal component scores were then plotted to better understand the relationship among each phenotypic variability component. The loadings from the PCA performed in the normative reference dataset were used to project new observations/patients into the PC-rotated coordinate system. The final mean-centered variance-standardized lower dimensional map, which contains the most information from our original variables, is the core of our approach.

### 2.4. Normative Model

Although the principal component analysis captures the variability of the autistic behaviors and its modifiers, it is still insufficient to address heterogeneity in autism at an individual level. To analyze the heterogeneity in the reference cohort while enabling predictions at the individual subject level, Gaussian Modeling was used to derive a normative model that captures the phenotypic variation in the reference sample by fitting a multivariate normal density to the PCA-derived coordinates. A special direction on the lower dimensional map, from all negative value corners to all positive corners, was defined as a gradient of general functionality. The probability of being less functional in this direction is calculated for any individual using a simple univariate normal density since, along this diagonal direction, the lower dimensional multivariate normal density is spherically symmetrical and reduces itself to the univariate case. The probability quantile calculated along the aforementioned “special” axis defines a relative Multidimensional General Functionality Score (MGFS) (details available as Appendix A).

### 2.5. Multidimensional General Functionality Score (MGFS)

In the phenotypic variance space, each patient is represented by a point at the lower dimensional coordinate system of *z*-score transformed PCs. A directed diagonal axis, which reflects general functionality, was defined from all negative octants to all positive octants. In order to estimate grades of general functionality, we calculated a 1-dimensional projection of an individual into the aforementioned axis. This projected point is the input variable for an accumulated probability integration based on the Gaussian modeling. The integrated quantile is then multiplied by 10 to yield the proposed Multidimensional General Functionality Score (MGFS) which represents the overall functionality and could be interpreted as the “level of support needs” when compared to the autistic reference sample. Expanded derivations and details on calculating the scores can be found in the Appendix A.

### 2.6. Statistical and Computational Analysis

Multivariate statistical methods and graphics were implemented using R statistical language (v4.0.3) scripts. The PCA implementation used was prcomp with all default parameters set to TRUE. Excel/LibreOffice .xls files are freely available as Appendix A to allow one to take advantage of the proposed normative approach for autistic individuals.

### 2.7. Overview of Steps

Figure 1 shows an overview of the 3 main phases of our approach: the application of PCA to the SSC sample, normative modeling, and the personalized evaluation strategy.

## 3. Results

In the following sections, we describe results obtained by the proposed approach according to the specific objectives: 1—to find principal components of phenotypic heterogeneity and evaluate their stability; 2—to construct the proposed scores and compare the Multidimensional General Functionality Score with severity scores used as a standard in clinical practice; and 3—to show how the proposed scores can be used in a clinical sample study.

### 3.1. Principal Component Analysis

We conducted a PCA on the subscales of VABS, ADI-R, and CBCL, as well as on instruments measuring IQ, to capture the multivariate components of phenotypic variability (Table 1 and Appendix A). Loading the contribution of each principal component (PC) showed that the first principal component (PC1) has a greater correlation (>0.65) with all VABS domain scores, total IQ, and ADI-R Socialization; the second principal component (PC2) has a greater correlation (>0.50) with CBCL Internalizing and Externalizing Problems and with ADI-R Restricted and Repetitive Behaviors; and the third principal component (PC3) has a greater correlation (>0.70) with ADI-R Communication. The same analysis was performed using a sample subset, and when using from 30% up to 80% of the SSC sample, all PCA results remain essentially the same (Appendix A).

Although PCs are linear combinations of original variables with no immediate pairing with instruments, the examination of individual top correlated variables with PCs yields some insights. Individuals located in the negative score octants on PC1 exhibit worse scores on VABS, Total IQ, and ADI-R Socialization. Individuals in the negative score octants on PC2 have worse scores on CBCL Internalizing and Externalizing Problems, and on ADI-R Restricted and Repetitive Behaviors. Individuals in the negative score octants in PC3 have worse scores in the ADI-R Communication.

Therefore, by sectorizing the three-dimensional projection of a PC coordinate system into eight octants, the association between the phenotypic presentation and PC values can be generalized as shown in Figure 2.

PC1 represents general and social functioning, ordering subjects from individuals with less general and social functionality to the individuals with better functionality. PC2 represents behavior problems representing a gradation of individuals with more behavior problems to individuals with fewer problems. Finally, the PC3 axis represents a gradation from individuals with more communication/language problems to individuals with fewer communication/language problems, assessed by the ADI-R subitem.

### 3.2. Quantifying Phenotype Heterogeneity Using a Multidimensional General Functionality Score

A general direction, from less general functionality and clinical presentation, which goes from octant VII’s vertices (all negative components) to octant I’s vertices (all positive components) was observed.

Along with patient visualization on a map endowed with natural clinical interpretation, this representation also allows normative modeling. The phenotypic heterogeneity contributing to severity embedded in the clinical reference sample was modeled by a probability density function and its quantiles provide a simple way to assess prospective patients’ general functionality. The modeled fraction of “worst of” patients multiplied by 10 returns a score for a given prospective individual that we named “Multidimensional general Functionality Score” (MGFS). The MGFS is a relative general functionality gradient from zero to ten. In contrast to standard quantiles used in growth charts, which are two-sided, the proposed MGFS score is one-sided since it assumes all negative and all positive octants as the less and best possible functionality, respectively, increasing along this direction (Figure 3). The distribution of MGFS calculated for the SSC subjects is essentially uniform, meaning that the proposed score increases approximately linearly along the diagonal axis and with the original total scores (Figure 4).

The ADOS-2 standardized Calibrated Severity Scores (CSSs) are a commonly used metric to assess ASD severity in clinical practice. The score was developed from the ADOS raw score and was modified to measure a patient’s autistic symptomatology severity with relative independence from individual characteristics such as language level and verbal IQ [12,13]. The ADOS-CSS provides an overall ASD severity measure regarding the core autistic behaviors and does not take into account the important role of other traits, such as internalizing/externalizing behaviors, IQ, functionality, and language levels, that contribute to characterize the autistic behaviors’ heterogeneity [20,21] and that have already been recognized in DSM-5 as modifiers.

The MGFS was developed to assign an overall functionality measure, accounting for both autistic behaviors and modifiers. Thus, as expected, our score shows no direct association with ADOS-2 standardized Calibrated Severity Score in the large SSC sample, but some relationships can be seen at the score’s boundaries (Figure 5). The heatmap shows the co-occurrence of MGFS (*y*-axis) and ADOS-2 CSS (*x*-axis) values received by the SSC probands, with green and blue colors indicating fewer probands with that specific co-occurrence, and purple and pink colors indicating more probands with that specific co-occurrence. As shown by the heatmap, among the probands who received an ADOS-2 CSS indicating the lowest severity level (four or five), the majority also received an MGFS indicating the highest general functionality level (nine or ten), while almost none received an MGFS indicating the lowest general functionality (0–2). Among the probands who received an ADOS-2 CSS indicating the highest severity level (ten), the majority also received an MGFS indicating the lowest general functionality level (one), and few received an MGFS indicating the highest functionality level (9–10).

### 3.3. Case Study

We present the suggested score usage on a subsample of autistic individuals [38], to show the general overview giving an idea of the localization of this sample according to the reference population. The phenotypic heterogeneity three-dimensional space was obtained using the SSC normative sample, but the main advantage of our approach resides in its translational power to any similarly obtained autistic individual. As our local sample has individuals assessed with equivalent standardized instruments, they can be mapped in the PCA-derived normative space using the same rotation eigenvalue built rotation matrix and normalization coefficients. Therefore, we assigned our 27 autistic individuals on the defined phenotypic heterogeneity map (Figure 6).

In addition to visualizing the individuals on the phenotypic heterogeneity space and empirically analyzing the relationships between their position and the individual’s clinical presentation, we quantified the position of each individual in the MGFS. In Figure 6, we can observe that compared to the reference map, most individuals are in a position that reflects the lowest general and social functionality, represented by a negative value on PC1, and more behavioral disturbance, represented by a negative value on PC2, but with fewer communication problems represented by a positive value on PC3 (the yellow octant in Figure 2). Based on these patterns, indicating the need for more support in two different domains, we would expect their MGFS to be less than or equal to five. In order to clarify how the phenotypic space and proposed scores can be used, we highlight the comparison between some individuals (Table 2). For instance, individuals ID-78, ID-31, and ID-33 are all located in the octant (−,−,+) with an MGFS less than or equal to five. In contrast, individuals such as ID-29 and ID-56, despite showing similar patterns, have MGFS values greater than five. Moreover, even among cases with MGFS values below five, their positions in the phenotypic space often fall within different octants. These results underscore the utility of the proposed map in delineating clinical homogeneity, accounting for factors that could otherwise compromise the reliability or reproducibility of clinical trial outcomes.

The proposed score can be used not only to compare the overall functionality and levels of support needed between two or more individuals but also to follow clinical changes in the same individual across different time points. This allows for a dynamic assessment of an individual’s progress or regression over time, offering insights into patterns of improvement, stability, or decline (Figure 7). Such evaluations are essential for tailoring interventions, monitoring treatment outcomes, and better understanding individual trajectories within the context of phenotypic variability. 

## 4. Discussion

Principal components, which explained up to 70% of the variability of an autistic reference sample, were used to create a map of phenotypic heterogeneity components contributing to the “highest support needed” to “lowest support needed” clinical spectrum. Based on normative modeling using the principal components of phenotypic heterogeneity, any patient evaluated by the phenotypic scales used can receive a score. To our knowledge, this is the first study that proposed a systematic way to combine: (i) quantitative access to the phenotype variability to assess how autism impacts a person’s life in a more comprehensive way that addresses the autistic behaviors as well as other influential aspects of an autistic person’s life; and (ii) normative modeling methods, thus allowing one to map and quantify an individual’s clinical heterogeneity in a defined normative space.

To establish the diagnosis of ASD, symptoms of both the Restricted Repetitive Behavior and social–communication domains are required to be present and allow individual variation in the quality and quantity of specific RRBs and social–communication deficits. In addition to the variability in the presentation of the main symptoms of ASD, it is recognized that affected individuals also vary in relation to non-ASD symptoms, such as cognitive ability, expressive language ability, onset patterns and comorbid psychopathology [45,46]. In addition to contributing to clinical heterogeneity, language, IQ, emotional regulation, and functionality (accessed by VABS and CBCL) contribute to different ASD trajectories. To create a score of characteristics recurrently associated with clinical heterogeneity and different ASD trajectories, first we sought to use PCA to reduce the dimensionality space. Using the established criteria, three PCs were selected, representing the important components that contribute to overall functionality and need for support. It is important to clarify that we are not discussing the diagnostic dimensions.

PC1 was correlated with total IQ, social skills (measured by ADI-R), and social, daily living, and communication skills measured by the VABS scores. The first component reflects findings from the literature indicating that IQ and adaptive behavior are correlated with socialization scores and interfere with social functioning [25,26,47]. Interestingly, VABS’s weight in this component is corroborated by a longitudinal study that has shown the important role of adaptive functioning in delineating ASD cases with different developmental trajectories [27,48]. Despite the fact that Intellectual Disability (ID) estimates in the ASD population have progressively decreased in recent decades to one-third of individuals with ASD, with 25% and 44% of individuals estimated in the ID/borderline and medium/above-average ranges, respectively [49], a longitudinal study indicated low IQ as a factor of poor prognosis [7,45,50]. Nonetheless, a higher IQ can be a necessary but insufficient condition for positive results in areas of functioning, such as relationships, employment, and independence [45,51,52]. Tillmann et al. (2019) [53] tested unique predictors of adaptive functioning as measured by the VABS and the discrepancy between IQ and adaptive functioning in ASD, and showed that socio-communicative symptoms, but not sensory/repetitive symptoms or concomitant psychiatric symptoms (anxiety, depression, and ADHD), are associated with less adaptive functioning and greater discrepancies in adaptive capacity.

Impairments in reciprocal social behavior, a key component of early social competence, clinically define autism spectrum disorder (ASD) [54]. While IQ captures the full range of intellectual functioning, assessments of adaptive functioning target the skills needed to perform daily activities and function independently, where a criterion level of intellectual functioning allows for the attainment of a certain level of adaptive functioning and increases in FSIQ are associated with improvements in adaptive skills and social responsiveness [4,55]. Kanne et al., 2011 [26] reaffirm the notion that adaptive deficits are a major challenge for intellectually capable individuals with ASD and, despite having a solid repertoire of verbal and nonverbal processing skills, these individuals have an even greater challenge in functionally applying their own strengths in everyday contexts, particularly in the areas of receptive and expressive communication, personal skills of daily living, and all three subdomains of socialization (interpersonal, play/leisure, and coping). That is, these individuals do not use their cognitive skills properly in order to improve their adaptive skills, especially social [56,57]. Stronger associations were observed between adaptive behavior and autism symptoms as reported by parents (i.e., ADI-R and SRS), particularly between adaptive socialization skills and social communication impairments. Thus, strong associations with VABS suggest that those individuals with more significant early social deficits have more challenging current adaptive levels [26,58].

The second component refers to internalizing and externalizing behaviors and Restricted Repetitive Behavior (RRB). RRB was considered a separate dimension of social communication before and after the DSM-5 classification; therefore, it is expected to be more correlated with a different PC than sociability. Psychiatric and medical comorbidities are present in about 70% of individuals with ASD [59], with anxiety disorders being the most common, affecting about 40% of children and adolescents with ASD [60]. In addition to anxiety, depression is also highly prevalent across the lifespan, occurring in 17% to 70% of individuals with ASD [61]. Emotional dysregulation is a risk factor for anxiety in ASD [62]. Georgiades et al. (2011) [19] examined the phenotypic overlap between core diagnostic features and emotional/behavioral problems in ASD using PCA. As noted, PCs associated with emotional behavior were not correlated with children’s intellectual, functional, adaptive, and structural language skills. Individuals with Intellectual Disability are at increased risk for behavioral problems, but children with ID have been shown to be at increased risk for only a few behavioral problems, such as self-injury and abnormal eating behaviors, validating that they represent distinct components [63].

Considering that we are seeking to understand and manage the multidimensional spectrum of autism and quantify its heterogeneity, the relationship between RRB and emotional dysregulation has already been observed in the literature. It has been suggested that the lack of flexibility and RRB may contribute to emotional dysregulation in children with ASD [64,65]. It has also been suggested that autistic individuals may sustain hyperfocus on upsetting past events and that there is a correlation between core symptoms of ASD and anger rumination [66,67]. In these and other studies, authors have discussed how RRBs and behavioral problems interact, contributing to ASD trajectories [68,69,70]. Children with ASD who exhibit high (versus low) restricted and repetitive behavior challenges appear to be at risk for greater emotion dysregulation [71]. Lower adaptive behaviors and repetitive behaviors (RRBs) correlated with higher externalizing behaviors. Lower cognitive abilities and higher RRBs in childhood predicted problems with externalizing behaviors in adolescence, i.e., the greater the impairment in cognitive ability and adaptive skills, the greater the occurrence of RRBs and externalizing behaviors in early childhood [72,73].

The third and final component was represented by ADI-R Communication. Considering that in the DSM-5 some communication items were included in the sociability dimension and others in the repetitive and stereotyped behaviors dimension, it may seem that our findings disagree with the two-dimensional model [74]. However, the two-dimensional model proposed in the DSM-5 is for diagnosis and, as already discussed, language ability is a modifier, since it is not necessary for diagnosis, although it interferes with heterogeneity [75]. Our scores are not for diagnosis, but to characterize heterogeneity. We suggest that what is represented in PC3, which is not represented in the other components, is precisely the language ability part of communication. Previous work seeking to create ASD ontologies mapped 27% of the items in the ADI-R specifically to language ability [76]. Additionally, the ADI-R scoring system has a coded communication subdomain for children at all language levels: gesture and play. Other subdomains are considered separately for children with sentences (verbal) and children with single words or no words (nonverbal) [40]. For verbal children, in addition to gesture and play, language ability/skills are also assessed. In the SSC, 88% are verbal and 12% are nonverbal, so language ability was assessed and contributed to finding components in the majority of cases in normative SSC samples. Language skills are not necessary for diagnosis, but 30% of children with ASD remain minimally verbal upon entering school [77,78,79]. A recent study found that lower adaptive outcomes, higher IQ (measured over time), and language ability in childhood tend to predict autonomy outcomes in adulthood [45,80].

Other studies are moving toward using multiple measures to construct more comprehensive profiles, such as including measures of intellectual functioning, emotional regulation, and adaptive behavior in understanding ASD heterogeneity [25,81,82,83], but we propose using these measures to create a heterogeneity score. Because we propose assessing clinical heterogeneity for a specific individual, we compare the MGFS to the CSS score, a modified ADOS score for assessing severity. Interestingly, the majority of SSC probands who received an MGFS of ten also received an ADOS-2 CSS of six, indicating a moderate level of severity. This result exemplifies that focusing on measures that were conceptualized and designed to diagnose autism can be problematic because these measures were designed with the primary goal of differentiating ASD from non-ASD, rather than describing clinical components that could contribute to ASD heterogeneity and trajectories. Different studies have shown that language ability, IQ, and VABS scores, rather than the core symptoms of ASD, exhibit the greatest variability and have the most significant impact on ASD trajectories [22,31,32,33,82]. For an adequate description of functional impairment, it is necessary to go beyond the ADOS-2 CSS to include information on language, adaptive functioning, and other behavioral characteristics. The results of this study add to the existing literature on ASD by emphasizing the importance of assessing general emotional/behavioral problems, functioning, IQ, and language ability together with core diagnostic symptoms in children with ASD to characterize clinical heterogeneity. The mapping proposed here allows clinical phenotypes to be dissected along the most relevant axes of variation. This approach allows us to map the variability across different domains of functioning and compare individuals in terms of this variability, aligning with the framework proposed by normative model studies [37]. By doing so, we developed a map that positions patients with ASD, enabling the assessment of their phenotypic relationships. This facilitates a more detailed characterization of individual patients and allows for standardized comparisons across cases.

The supposed translational success of this normative model is based on the power of the normative sample to serve as a relevant reference population. The SFARI SSC database is a renowned collective effort to centralize and distribute phenotype and genotype/genomic information [39]. It is important to discuss some limitations of the present study. Although the emphasis was on selecting measures that represent distinct traits, rather than specific instruments or scales, we acknowledge that the selected input variables used in the PCA analysis do not represent the single/best selection or the full range of the ASD-related phenotype. Additional measures that encompass motor skills or language are being considered in follow-up work. The scales were used with age-normative scores, allowing for the construction of a map of phenotypic heterogeneity, but studies with repeated measures over time may add information on temporal stability and trait selection for map construction. Repeated measures would also be very important to delineate trajectories. Input data for PCA were collected using parent reports (ADI-R, CBCL, and VABS) and are therefore subject to bias. Another limitation is that the normative sample (SSC) consists of simplex families and does not necessarily represent the full genetic architecture of ASD.

## 5. Conclusions

This work offers a series of contributions to research and practice. First, from the research point of view, we propose a model to observe a patient considering the heterogeneity of clinical phenotypic relationships, and provide patient placement in an ASD heterogeneity map, allowing reliable case comparisons even among time-points of the same individual. Moreover, the approach proposed here presents an important differential: the idea of looking at a single subject. This can be important for treatment proposals and evaluation since individuals with ASD often differ in response to treatment. These findings may facilitate the development of more effective therapeutic strategies to optimize long-term results for individuals with ASD in a personalized medicine view, as well as sample selection for clinical and biomarker studies.

## Figures and Tables

**Figure 1 brainsci-14-01254-f001:**
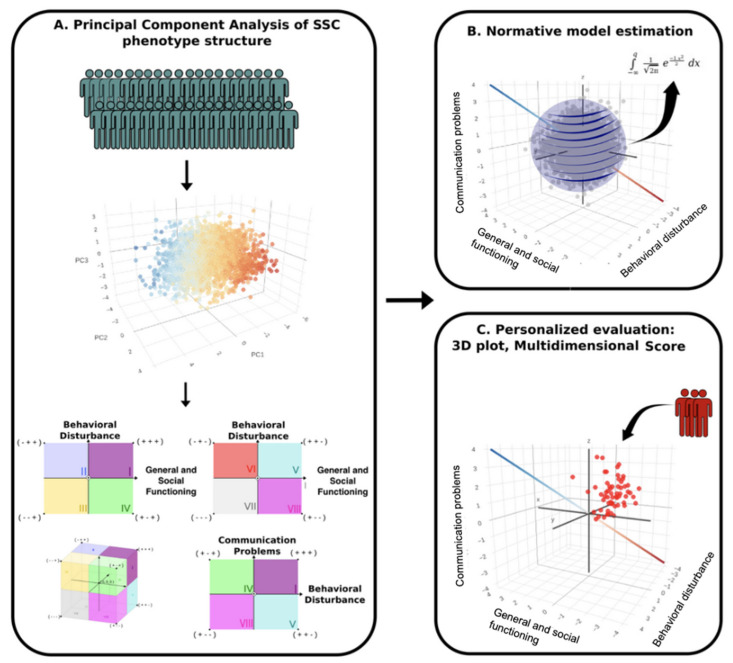
Overview of the method. (**A**) Phenotypic variability map construction, based on the SSC autistic individuals’ coordinates in three principal components. Each sector of the principal component coordinate system has a clinical interpretation, resulting in three axes of phenotypic variability: (**B**) “General and Social Functioning”, “Behavioral Disturbance”, and “Communication/language Problems”. Gaussian modeling was used to derive a normative model that captures the phenotypic variation in the reference sample by fitting a multivariate normal density to the PCA-derived coordinates, concerning a special direction on the 3-dimensional map defined as a gradient direction of clinical presentation (**C**). Any new patient can be mapped in the 3D space endowed with clinical interpretation and receive a “Multidimensional General Functionality Score”.

**Figure 2 brainsci-14-01254-f002:**
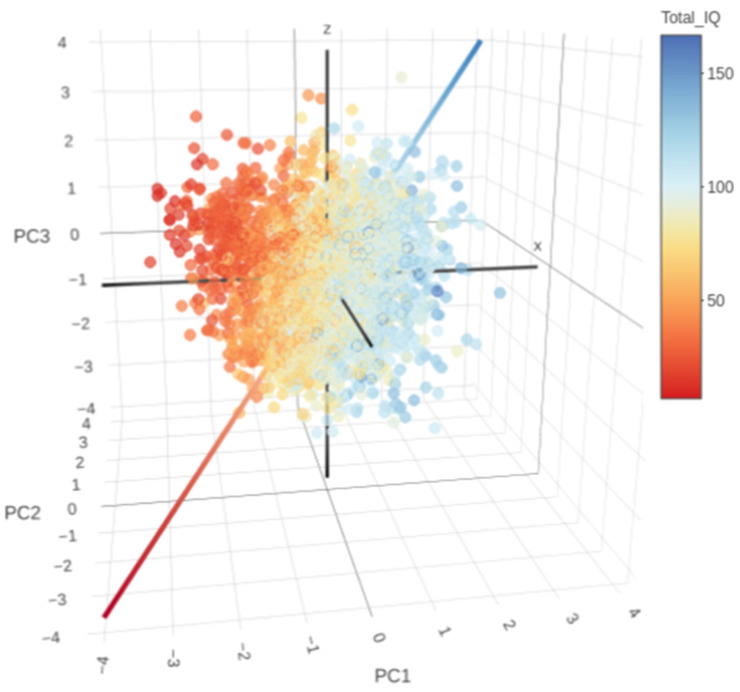
Relationship between Total Intelligence Quotient (IQ) and the first three principal components coordinate system, normalized to z-scores. The line is directed from worst (all negative, red) to better (all positive, blue) and crosses the origin.

**Figure 3 brainsci-14-01254-f003:**
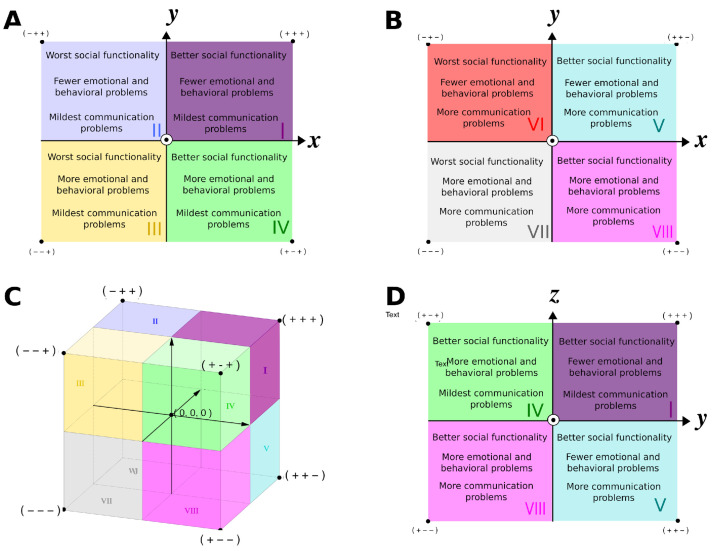
Clinical and conceptual implications of dimensional representation of autistic individuals. The tridimensional space proposed to holistically represent individuals is divided into eight octants, labeled from I to VIII (**C**). Schematically, the *x*-, *y*-, and *z*-axis embed principal components one, two, and three, respectively. Two-dimensional views of the space are shown for clarity (**A**,**B**,**D**) along with octant clinical interpretation (text inside). Each octant corner indicated with “+” or “−” signals qualitative better or worse clinical status for the three dimensions. The circled dot at the origin represents an axis directed towards the outside of the plane shown and *z*-axis positive and negative octants (**A**,**B**, respectively) are shown separately for clarity.

**Figure 4 brainsci-14-01254-f004:**
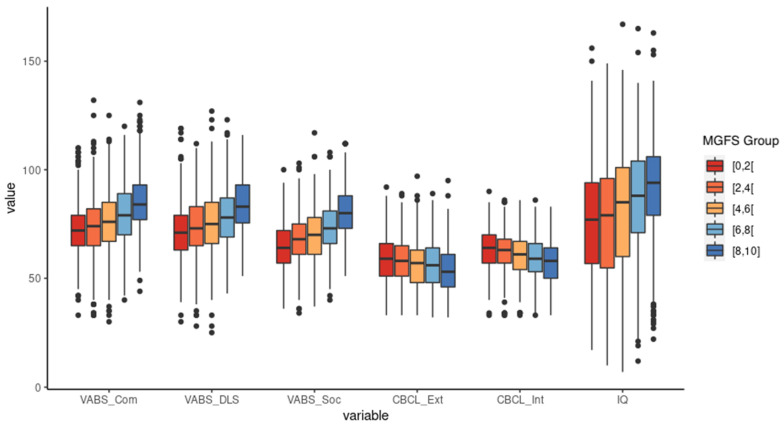
Boxplot of the original variables scores according to the MGFS. The *x*-axis shows the original variables used to construct the MGFS (except ADI-R subitems). Each color in the boxplot indicates a different range of MGFS, with the red boxplot (0–1.9) indicating the group that needs more support and the blue boxplot (MGFS 8–10) indicating the group that needs less support.

**Figure 5 brainsci-14-01254-f005:**
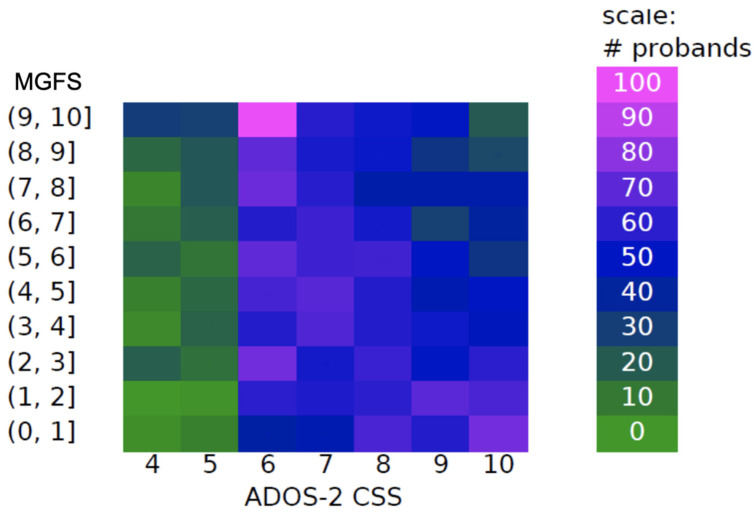
Relationship between MGFS and ADOS-2 Calibrated Severity Scores (CSSs). Heatmap depicting the distribution of probands based on the Multidimensional General Functionality Score (MGFS) and ADOS-2 calibrated severity scores (CSSs). The *y*-axis represents MGFS ranges, while the *x*-axis represents ADOS-2 CSSs. The color intensity indicates the number of probands in each cell, as shown in the scale bar on the right. Green represents low counts, transitioning to blue, purple, and pink for higher counts, with pink indicating the highest number of probands (100).

**Figure 6 brainsci-14-01254-f006:**
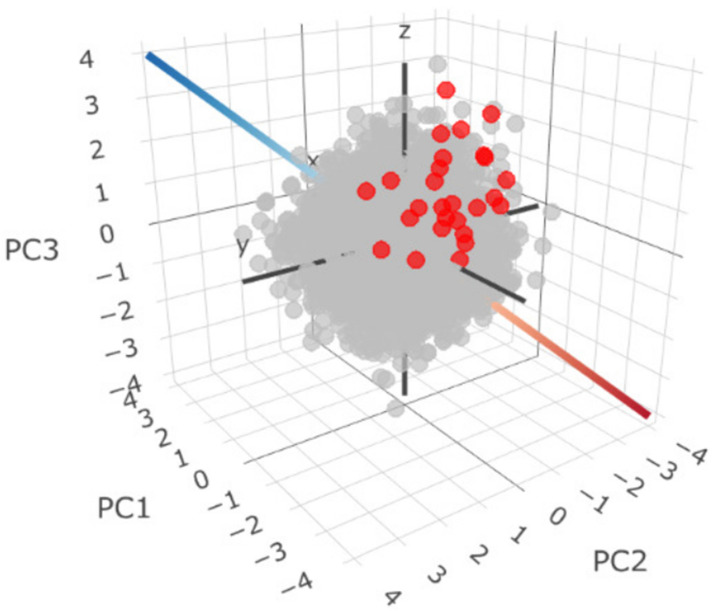
Visualization of case study individuals on the map of phenotypic heterogeneity under normative modeling. The case study individuals are shown in red. The principal components account for 73% of total variance distributed as PC1 (39%), PC2 (18%), and PC3 (15%).

**Figure 7 brainsci-14-01254-f007:**
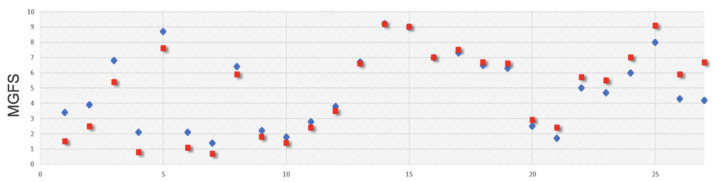
Longitudinal comparison of MGFS. This figure illustrates the Multidimensional General Functionality Score (MGFS) for each of the 27 patients at two distinct time points. Blue points represent the MGFS values at baseline, while red points represent the values measured after 8 months. This comparison highlights changes in functionality over time for each patient.

**Table 1 brainsci-14-01254-t001:** Principal component analysis (PCA) of SSC reference sample. The model considered the first three principal components, capturing approximately 70% of the sample variability. Pearson correlations between these components and the original input variables are presented, with values greater than 0.50 highlighted in bold. A detailed table showing correlations for all nine components is provided in Appendix A.

Component	1	2	3
Eigenvalue	3.49	1.65	1.39
Explained Variance (%)	39	18	15
Cumulative Explained Variance (%)	39	57	72
**Correlations**			
VABS Communication	**0.89**	−0.16	0.17
VABS Socialization	**0.87**	0.04	0.12
VABS Daily Living Skills	**0.85**	−0.13	0.19
Total IQ	**0.78**	−0.29	0.12
ADI-R Socialization	**−0.69**	−0.18	0.43
ADI-R Communication	−0.31	−0.34	**0.73**
ADI-R Restricted and Repetitive Behavior	−0.17	**−0.52**	0.45
CBCL Internalizing Problems	−0.04	**−0.78**	−0.39
CBCL Externalizing Problems	−0.14	**−0.71**	−0.48

**Table 2 brainsci-14-01254-t002:** Participant characteristics and assessment scores. The table displays individual participant IDs, scores from the Vineland Adaptive Behavior Scales (VABS) for Communication (Com), Socialization (Soc), and Daily Living Skills (DLS); Autism Diagnostic Interview (ADI) scores for Social (Soc), Communication (Com), and Restricted, Repetitive Behaviors (RRB); Total IQ scores; Child Behavior Checklist (CBCL) Internalizing (Int) and Externalizing (Ext) problem scores; Scores for the first three components (PC1, PC2, PC3) from PCA; categorical octant classifications based on PCA; and the Multidimensional General Functionality Score (MGFS).

IDs	VABS Com	VABS Soc	VABS DLS	ADI Soc	ADI Com	ADI RRB	Total IQ	CBCL Int	CBCL Ext	PC1	PC2	PC3	Octant	MGFS
ID-29	50	51	54	28	7	5	54	72	56	−1.55	0.36	1.87	−,+,+	6.5
ID-56	49	51	52	23	13	4	67	59	56	−1.41	0.87	1.14	−,+,+	6.4
ID-77	47	51	45	24	19	3	70	63	62	−1.71	0.29	0.88	−,+,+	3.8
ID-39	31	45	32	24	13	10	49	66	51	−2.48	0.29	0.80	−,+,+	2.1
ID-33	47	54	35	28	16	5	49	77	77	−2.18	−0.99	1.82	−,−,+	2.2
ID-78	47	57	53	23	20	6	59	65	67	−1.60	−0.38	0.56	−,−,+	2.1
ID-31	33	49	31	25	14	9	49	72	67	−2.49	−0.59	1.42	−,−,+	1.7
ID-47	49	61	49	24	20	8	66	60	64	−1.56	−0.31	−0.02	−,−,−	1.4

## Data Availability

Restrictions apply to the availability of the data. Data were obtained from the SFARI database and are available from (https://www.sfari.org/, access on 11 December 2024) with the permission of the SFARI foundation.

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
