# Peer review of "A Normative Model Representing Autistic Individuals Amidst Autism Spectrum Phenotypic Heterogeneity"

_brainsci, 2024, doi:10.3390/brainsci14121254_

Round 1
Reviewer 1 Report
Comments and Suggestions for Authors
Thank you for asking me to review this interesting manuscript. The authors have taken a principal components approach to describing high dimensional phenotypes in people with ASD. Their approach also generates normative model that allows for a score based on multiple dimensions that could be useful for describing ASD phenotypes. This information could be useful for defining appropriate interventions for specific patients with the view that the most appropriate interventions can be emphasized. This is a well-conceived and well-constructed study. I only have minor concerns.
1. There is a heatmap referred to on Pg 10 but I did not see a heatmap in the article.
2. On Pg 12, Fig 4 should say Fig 5
3. Is age included as a variable in the PCA and if so, did it simply fall into PCs >3
4. I wonder if the authors could be a bit bolder about potential uses of the score. I agree that it defines heterogeneity, but it could do more than that. For example, do the authors think that the scores are stable over time in individuals or could change in score be a strong marker of therapeutic success. In addition, is there predictive value in the score i.e. does a score at T1 predict a score at a subsequent time T2 with reliability? If so, this score could be very useful in the clinic for predicting clinical course that could be very useful for families.
5. I do not feel that I am appropriately qualified to evaluate the detailed mathematical aspect of the manuscript. My comments are based on the idea that there are no major flaws in this domain.
Author Response
Editor's comments: Thank you for asking me to review this interesting manuscript. The authors have taken a principal components approach to describing high dimensional phenotypes in people with ASD. Their approach also generates normative model that allows for a score based on multiple dimensions that could be useful for describing ASD phenotypes. This information could be useful for defining appropriate interventions for specific patients with the view that the most appropriate interventions can be emphasized. This is a well-conceived and well-constructed study. I only have minor concerns.
- There is a heatmap referred to on Pg 10 but I did not see a heatmap in the article.
- On Pg 12, Fig 4 should say Fig 5
- Is age included as a variable in the PCA and if so, did it simply fall into PCs >3
- I wonder if the authors could be a bit bolder about potential uses of the score. I agree that it defines heterogeneity, but it could do more than that. For example, do the authors think that the scores are stable over time in individuals or could change in score be a strong marker of therapeutic success. In addition, is there predictive value in the score i.e. does a score at T1 predict a score at a subsequent time T2 with reliability? If so, this score could be very useful in the clinic for predicting clinical course that could be very useful for families.
- I do not feel that I am appropriately qualified to evaluate the detailed mathematical aspect of the manuscript. My comments are based on the idea that there are no major flaws in this domain.
Response to Reviewer 1
1. Missing heatmap (Pg 10): We apologize for the oversight. We have now included the heatmap in the revised manuscript and ensured it is appropriately referenced in the text as Fig 5 in Pg 11.
2. Incorrect figure reference (Pg 12): The incorrect reference to Fig. 4 has been corrected to Fig. 6, after the heatmap inclusion.
3. Age as a variable in PCA: Age was not included as a variable in the PCA analysis due to its unavailability in the SSC cohort. However, we acknowledge the critical importance of considering children's age when evaluating clinical trajectories and the complexity of phenotypic presentations.
4. Potential uses of the score: We appreciate the reviewer’s insightful question regarding the potential uses of the score and its clinical relevance. While the current study primarily focused on defining heterogeneity at a single time point, we agree that exploring the utility of the score over time is an important next step.
As suggested, we included an initial demonstration using clinical cases evaluated at two time points to test whether the derived scores effectively capture changes in the same patient over time. This analysis is presented in Figure 7 (pages 13 and 14), where the blue dots represent the MGFS at baseline and the red dots represent the scores after 8 months.
Importantly, we emphasize that our current dataset does not include sufficient longitudinal data points to fully assess the predictive utility of the score over time (i.e., using the score at time 1 to predict the clinical trajectory at time 2) Instead, our demonstration focuses on using the score in repeated measures to evaluate a patient’s global clinical evolution. Nevertheless, changes in the score over time could provide a valuable tool for predicting clinical trajectories, offering families and clinicians a quantitative means to anticipate future needs or evaluate the potential benefits of targeted interventions.

Reviewer 2 Report
Comments and Suggestions for Authors
the manuscript "normative model representing autistic individuals amidst autism spectrum phenotypic heterogeneity" found 3 dimensions, hence 8 subtypes of ASD. This is not surprising given the work by Happe and colleagues (e.g., Time to give up on a single explanation for autism | Nature Neuroscience)
or that of Leif Ekblad, finding multiple phemotypes (12 dimensions), see Autism, Personality, and Human Diversity: Defining Neurodiversity in an Iterative Process Using Aspie Quiz - Leif Ekblad, 2013
how many of the over 2000 children (2-18 y) from the SSC fall into each of the octants?
your novelty is that given the questionnaires, one can "predict" the trajectory of the person? That is depending on how much a person loads on communication, emotion and social, the outlook for mastering our society is either bad, medium or good? Is that the rational behind your model? If so, the article could be strengthened by suggesting targeted training / intervention to assist persons with a shortcoming on one (two or all three) of the dimensions.
Bit unclear what the rational of the MGFS is, knowing that a person scores high on social / communication / emotion is already practice - hence the use of more than one questionnaire. Likely I have missed something.
minor:
line 114 than can impacts <- remove can
line 190 your <- should likely be our
line 191 . missing
the equations might need to be inserted as figures as they make no sense as they appear in print now
Author Response
Reviewer 2
The manuscript "normative model representing autistic individuals amidst autism spectrum phenotypic heterogeneity" found 3 dimensions, hence 8 subtypes of ASD. This is not surprising given the work by Happe and colleagues (e.g., Time to give up on a single explanation for autism | Nature Neuroscience)
or that of Leif Ekblad, finding multiple phenotypes (12 dimensions), see Autism, Personality, and Human Diversity: Defining Neurodiversity in an Iterative Process Using Aspie Quiz - Leif Ekblad, 2013
how many of the over 2000 children (2-18 y) from the SSC fall into each of the octants?
your novelty is that given the questionnaires, one can "predict" the trajectory of the person? That is depending on how much a person loads on communication, emotion and social, the outlook for mastering our society is either bad, medium or good? Is that the rational behind your model? If so, the article could be strengthened by suggesting targeted training / intervention to assist persons with a shortcoming on one (two or all three) of the dimensions.
Bit unclear what the rational of the MGFS is, knowing that a person scores high on social / communication / emotion is already practice - hence the use of more than one questionnaire. Likely I have missed something.
minor:
line 114 than can impacts <- remove can
line 190 your <- should likely be our
line 191 . missing
the equations might need to be inserted as figures as they make no sense as they appear in print now.
Response to Reviewer 2
We sincerely thank you for your thoughtful comments and for providing references to related works that align with our study. Your insights have helped us refine the manuscript and better articulate the novelty and implications of our findings. Below, we address your points in detail:
Minor revisions: All minor revisions were corrected in the text and equations reformatted as figures in the revised manuscript.
On the Subtypes of ASD and Their Distribution Across Octants: We appreciate your observation regarding prior works identifying multiple subtypes within the autism spectrum, such as Happe and colleagues’ dimensional approach and Ekblad’s work on 12 dimensions and we added these references in our discussion, although we do not define the patient's position in the octants as subtypes.
On the Rationale Behind the MGFS: The MGFS aims to provide a single, composite score that integrates the individual’s position across all three key dimensions of autism clinical presentation, offering a quantitative measure of overall functionality. While it is true that high scores on individual dimensions can already provide clinical insights, the MGFS allows for standardized comparisons across individuals and groups. For instance, it enables clinicians and researchers to track changes over time or evaluate interventions in a more systematic manner. Additionally, the MGFS provides a bridge between the dimensional model and practical clinical applications by summarizing complex phenotypic profiles into an easily interpretable number. To address your concern that this may not have been sufficiently clear, we have revised the manuscript to provide a more explicit explanation of the MGFS's rationale and practical utility, supported by examples.
On the Predictive Trajectory of the Model: You are correct that one of the key novelties of our model lies in its potential to provide insights into a person’s developmental trajectory. By positioning individuals within the phenotypic heterogeneity space, we can measure their relative strengths and challenges across the three dimensions. For instance, individuals with significant difficulties in more than one dimension may need more support to achieve social demands. We agree that this is an exciting and practical implication of our model and have clarified this point in the discussion.
We also included an initial demonstration using clinical cases evaluated at two time points to test whether the derived scores effectively capture changes in the same patient over time. This analysis is presented in Figure 7 (pages 13 and 14), where the blue dots represent the MGFS at baseline and the red dots represent the scores after 8 months.
Importantly, we emphasize that our current dataset does not include sufficient longitudinal data points to fully assess the predictive utility of the score over time (i.e., using the score at time 1 to predict the clinical trajectory at time 2). Instead, our demonstration focuses on using the score in repeated measures to evaluate a patient’s global clinical evolution. Nevertheless, changes in the score over time could provide a valuable tool for predicting clinical trajectories, offering families and clinicians a quantitative means to anticipate future needs or evaluate the potential benefits of targeted interventions.
Your suggestion to use the model to propose targeted training or interventions is excellent and aligns with our long-term goals. While this falls outside the scope of the current study, we acknowledge the potential for the MGFS and the phenotypic map to inform personalized interventions.
